# Escaping Saddle Points in Constrained Optimization

**Aryan Mokhtari**
MIT
Cambridge, MA 02139
aryanm@mit.edu

**Asuman Ozdaglar**
MIT
Cambridge, MA 02139
asuman@mit.edu

**Ali Jadbabaie**
MIT
Cambridge, MA 02139
jadbabai@mit.edu

## Abstract

In this paper, we study the problem of escaping from saddle points in smooth nonconvex optimization problems subject to a convex set $\mathcal{C}$. We propose a generic framework that yields convergence to a second-order stationary point of the problem, if the convex set $\mathcal{C}$ is simple for a quadratic objective function. Specifically, our results hold if one can find a $\rho$-approximate solution of a quadratic program subject to $\mathcal{C}$ in polynomial time, where $\rho < 1$ is a positive constant that depends on the structure of the set $\mathcal{C}$. Under this condition, we show that the sequence of iterates generated by the proposed framework reaches an $(\epsilon, \gamma)$-second order stationary point (SOSP) in at most $\mathcal{O}(\max\{\epsilon^{-2}, \rho^{-3}\gamma^{-3}\})$ iterations. We further characterize the overall complexity of reaching an SOSP when the convex set $\mathcal{C}$ can be written as a set of quadratic constraints and the objective function Hessian has a specific structure over the convex set $\mathcal{C}$. Finally, we extend our results to the stochastic setting and characterize the number of stochastic gradient and Hessian evaluations to reach an $(\epsilon, \gamma)$-SOSP.

## 1 Introduction

There has been a recent revival of interest in non-convex optimization, due to obvious applications in machine learning. While the modern history of the subject goes back six or seven decades, the recent attention to the topic stems from new applications as well as availability of modern analytical and computational tools, providing a new perspective on classical problems. Following this trend, in this paper we focus on the problem of minimizing a smooth nonconvex function over a convex set as follows:

$$\text{minimize } f(\mathbf{x}), \qquad \text{subject to } \mathbf{x} \in \mathcal{C}, \tag{1}$$

where $\mathbf{x} \in \mathbb{R}^d$ is the decision variable, $\mathcal{C} \subset \mathbb{R}^d$ is a closed convex set, and $f : \mathbb{R}^d \to \mathbb{R}$ is a twice continuously differentiable function over $\mathcal{C}$. It is well known that finding a global minimum of Problem (1) is hard. Equally well-known is the fact that for certain nonconvex problems, all local minimizers are global. These include, for example, matrix completion [24], phase retrieval [42], and dictionary learning [43]. For such problems, finding a global minimum of Problem (1) reduces to the problem of finding one of its local minima.

Given the well-known hardness results in finding stationary points, recent focus has shifted in characterizing approximate stationary points. When the objective function $f$ is convex, finding an $\epsilon$-first-order stationary point is often sufficient since it leads to finding an approximate local (and hence global) minimum. However, in the nonconvex setting, even when the problem is unconstrained, i.e., $\mathcal{C} = \mathbb{R}^d$, convergence to a first-order stationary point (FOSP) is not enough as the critical point to which convergence is established might be a saddle point. It is therefore natural to look at higher order derivatives and search for a second-order stationary points. Indeed, under the assumption that all the saddle points are strict (formally defined later), in both unconstrained and constrained settings, convergence to a second-order stationary point (SOSP) implies convergence to a local minimum.

While convergence to an SOSP has been thoroughly investigated in the recent literature for the unconstrained setting, the overall complexity for the constrained setting has not been studied yet.

**Contributions.** Our main contribution is to propose a generic framework which generates a sequence of iterates converging to an approximate second-order stationary point for the constrained nonconvex problem in (1), when the convex set $\mathcal{C}$ has a specific structure that allows for approximate minimization of a quadratic loss over the feasible set. The proposed framework consists of two main stages: First, it utilizes first-order information to reach a first-order stationary point; next, it incorporates second-order information to escape from a stationary point if it is a local maximizer or a strict saddle point. We show that the proposed approach leads to an $(\epsilon, \gamma)$-second-order stationary point (SOSP) for Problem (1) (check Definition 1). The proposed approach utilizes advances in constant-factor optimization of nonconvex quadratic programs [46, 22, 44] that find a $\rho$-approximate solution over $\mathcal{C}$ in polynomial time, where $\rho$ is a positive constant smaller than 1 that depends on the structure of $\mathcal{C}$. When such approximate solution exists, the sequence of iterates generated by the proposed framework reaches an $(\epsilon, \gamma)$-SOSP of Problem (1) in at most $\mathcal{O}(\max\{\epsilon^{-2}, \rho^{-3}\gamma^{-3}\})$ iterations.

We show that quadratic constraints satisfy the required condition for the convex set $\mathcal{C}$ if the objective function Hessian $\nabla^2 f$ has a specific structure over the convex set $\mathcal{C}$ (formally described later). For this case, we show that it is possible to achieve an $(\epsilon, \gamma)$-SOSP after at most $\mathcal{O}(\max\{\tau\epsilon^{-2}, d^3 m^7 \gamma^{-3}\})$ arithmetic operations, where $d$ is the dimension of the problem and $\tau$ is the number of required arithmetic operations to solve a linear program over $\mathcal{C}$ or to project a point onto $\mathcal{C}$. We further extend our results to the stochastic setting and show that we can reach an $(\epsilon, \gamma)$-SOSP after computing at most $\mathcal{O}(\max\{\epsilon^{-4}, \epsilon^{-2}\rho^{-4}\gamma^{-4}, \rho^{-7}\gamma^{-7}\})$ stochastic gradients and $\mathcal{O}(\max\{\epsilon^{-2}\rho^{-3}\gamma^{-3}, \rho^{-5}\gamma^{-5}\})$ stochastic Hessians.

## 1.1 Related work

**Unconstrained case**. The rich literature on nonconvex optimization provides a plethora of algorithms for reaching stationary points of a smooth *unconstrained* minimization problem. Convergence to first-order stationary points (FOSP) has been widely studied for both deterministic [35, 1, 7–10] and stochastic settings [39, 38, 3, 32]. Stronger results which indicate convergence to an SOSP are also established. Numerical optimization methods such as trust-region methods [13, 19, 33] and cubic regularization algortihms [36, 11, 12] can reach an approximate second-order stationary point in a finite number of iterations; however, typically the computational complexity of each iteration could be relatively large due to the cost of solving trust-region or regularized cubic subproblems. Recently, a new line of research has emerged that focuses on the overall computational cost to achieve an SOSP. These results build on the idea of escaping from strict saddle points with perturbing the iterates by injecting a properly chosen noise [23, 29, 30], or by updating the iterates using the eigenvector corresponding to the smallest eigenvalue of the Hessian [7, 2, 45, 41, 1, 40, 37].

**Constrained case**. Asymptotic convergence to first-order and second-order stationary points for the constrained optimization problem in (1) has been studied in the numerical optimization community [6, 18, 21, 20]. Recently, finite-time analysis for convergence to an FOSP of the generic smooth constrained problem in (1) has received a lot of attention. In particular, [31] shows that the sequence of iterates generated by the update of Frank-Wolfe converges to an $\epsilon$-FOSP after $\mathcal{O}(\epsilon^{-2})$ iterations. The authors of [26] consider norm of gradient mapping as a measure of non-stationarity and show that the projected gradient method has the same complexity of $\mathcal{O}(\epsilon^{-2})$. Similar result for the accelerated projected gradient method is also shown [25]. Adaptive cubic regularization methods in [14–16] improve these results using second-order information and obtain an $\epsilon$-FOSP of Problem (1) after at most $\mathcal{O}(\epsilon^{-3/2})$ iterations. Finite time analysis for convergence to an SOSP has also been studied for linear constraints. In particular, [5] studies convergence to an SOSP of (1) when the set $\mathcal{C}$ is a linear constraint of the form $\mathbf{x} \geq 0$ and propose a trust region interior point method that obtains an $(\epsilon, \sqrt{\epsilon})$-SOSP in $\mathcal{O}(\epsilon^{-3/2})$ iterations. The work in [27] extends their results to the case that the objective function is potentially not differentiable or not twice differentiable on the boundary of the feasible region. The authors in [17] focus on the general convex constraint case and introduce a trust region algorithm that requires $\mathcal{O}(\epsilon^{-3})$ iterations to obtain an SOSP; however, each iteration of their proposed method requires access to the exact solution of a nonconvex quadratic program (finding its global minimum) which, in general, could be computationally prohibitive. To the best of our knowledge, our paper provides the first finite-time overall computational complexity analysis for reaching an SOSP of Problem (1).

## 2 Preliminaries and Definitions

In the case of *unconstrained* minimization of the objective function $f$, the first-order and second-order necessary conditions for a point $\mathbf{x}^*$ to be a local minimum of that are defined as $\nabla f(\mathbf{x}^*) = \mathbf{0}_d$ and $\nabla^2 f(\mathbf{x}^*) \succeq \mathbf{0}_{d \times d}$, respectively. If a point satisfies these conditions it is called a *second-order stationary point* (SOSP). If the second condition becomes strict, i.e., $\nabla^2 f(\mathbf{x}) \succ \mathbf{0}$, then we recover the sufficient conditions for a local minimum. However, to derive finite time convergence bounds for achieving an SOSP, these conditions should be relaxed. In other words, the goal should be to find an *approximate* SOSP where the approximation error can be arbitrarily small. For the case of unconstrained minimization, a point $\mathbf{x}^*$ is called an $(\epsilon, \gamma)$-second-order stationary point if it satisfies $\|\nabla f(\mathbf{x}^*)\| \leq \epsilon$ and $\nabla^2 f(\mathbf{x}^*) \succeq -\gamma \mathbf{I}_d$, where $\epsilon$ and $\gamma$ are arbitrary positive constants. To study the constrained setting, we first state the necessary conditions for a local minimum of Problem (1).

**Proposition 1** ([4]). *If $\mathbf{x}^* \in \mathcal{C}$ is a local minimum of the function $f$ over the convex set $\mathcal{C}$, then*

$$\nabla f(\mathbf{x}^*)^\top (\mathbf{x} - \mathbf{x}^*) \geq 0, \quad \text{for all } \mathbf{x} \in \mathcal{C}, \tag{2}$$

$$(\mathbf{x} - \mathbf{x}^*)^\top \nabla^2 f(\mathbf{x}^*)(\mathbf{x} - \mathbf{x}^*) \geq 0, \quad \text{for all } \mathbf{x} \in \mathcal{C} \ \text{ s.t. } \nabla f(\mathbf{x}^*)^\top (\mathbf{x} - \mathbf{x}^*) = 0. \tag{3}$$

The conditions in (2) and (3) are the first-order and second-order necessary optimality conditions, respectively. By making the inequality in (3) strict, i.e., $(\mathbf{x} - \mathbf{x}^*)^\top \nabla^2 f(\mathbf{x}^*)(\mathbf{x} - \mathbf{x}^*) > 0$, we recover the sufficient conditions for a local minimum when $\mathcal{C}$ is a polyhedral [4]. Further, if the inequality in (3) is replaced by $(\mathbf{x} - \mathbf{x}^*)^\top \nabla^2 f(\mathbf{x}^*)(\mathbf{x} - \mathbf{x}^*) \geq \delta \|\mathbf{x} - \mathbf{x}^*\|^2$ for some $\delta > 0$, we obtain the sufficient conditions for a local minimum of Problem (1) for any convex constraint $\mathcal{C}$; see [4]. If a point $\mathbf{x}^*$ satisfies the conditions in (2) and (3) it is an SOSP of Problem (1).

As in the unconstrained setting, the first-order and second-order optimality conditions may not be satisfied in finite number of iterations, and we focus on finding an approximate SOSP.

**Definition 1.** *Recall the twice continuously differentiable function $f : \mathbb{R}^d \to \mathbb{R}$ and the convex closed set $\mathcal{C} \subset \mathbb{R}^d$ introduced in Problem (1). We call $\mathbf{x}^* \in \mathcal{C}$ an $(\epsilon, \gamma)$-second order stationary point of Problem (1) if the following conditions are satisfied.*

$$\nabla f(\mathbf{x}^*)^\top (\mathbf{x} - \mathbf{x}^*) \geq -\epsilon, \quad \text{for all } \mathbf{x} \in \mathcal{C}, \tag{4}$$

$$(\mathbf{x} - \mathbf{x}^*)^\top \nabla^2 f(\mathbf{x}^*)(\mathbf{x} - \mathbf{x}^*) \geq -\gamma, \quad \text{for all } \mathbf{x} \in \mathcal{C} \ \text{ s.t. } \nabla f(\mathbf{x}^*)^\top (\mathbf{x} - \mathbf{x}^*) = 0. \tag{5}$$

*If a point only satisfies the first condition, we call it an $\epsilon$-first order stationary point.*

We further formally define strict saddle points for the constrained optimization problem in (1).

**Definition 2.** *A point $\mathbf{x}^* \in \mathcal{C}$ is a $\delta$-strict saddle point of Problem (1) if (i) for all $\mathbf{x} \in \mathcal{C}$ the condition $\nabla f(\mathbf{x}^*)^\top (\mathbf{x} - \mathbf{x}^*) \geq 0$ holds, and (ii) there exists a point $\mathbf{y}$ such that*

$$(\mathbf{y} - \mathbf{x}^*)^\top \nabla^2 f(\mathbf{x}^*)(\mathbf{y} - \mathbf{x}^*) < -\delta, \quad \mathbf{y} \in \mathcal{C} \text{ and } \nabla f(\mathbf{x}^*)^\top (\mathbf{y} - \mathbf{x}^*) = 0. \tag{6}$$

According to Definitions 1 and 2 if all saddle points are $\delta$-strict and $\gamma \leq \delta$, any $(\epsilon, \gamma)$-SOSP of Problem (1) is an approximate local minimum.

We emphasize that in this paper we do not assume that all saddles are strict to prove convergence to an SOSP. We formally defined strict saddles just to clarify that if all the saddles are strict then convergence to an approximate SOSP is equivalent to convergence to an approximation local minimum.

Our goal throughout the rest of the paper is to design an algorithm which finds an $(\epsilon, \gamma)$-SOSP of Problem (1). To do so, we first assume the following conditions are satisfied.

**Assumption 1.** *The gradients $\nabla f$ are $L$-Lipschitz continuous over the set $\mathcal{C}$, i.e., for any $\mathbf{x}, \tilde{\mathbf{x}} \in \mathcal{C}$,*

$$\|\nabla f(\mathbf{x}) - \nabla f(\tilde{\mathbf{x}})\| \leq L \|\mathbf{x} - \tilde{\mathbf{x}}\|. \tag{7}$$

**Assumption 2.** *The Hessians $\nabla^2 f$ are $M$-Lipschitz continuous over the set $\mathcal{C}$, i.e., for any $\mathbf{x}, \tilde{\mathbf{x}} \in \mathcal{C}$*

$$\|\nabla^2 f(\mathbf{x}) - \nabla^2 f(\tilde{\mathbf{x}})\| \leq M \|\mathbf{x} - \tilde{\mathbf{x}}\|. \tag{8}$$

**Assumption 3.** *The diameter of the compact convex set $\mathcal{C}$ is upper bounded by a constant $D$, i.e.,*

$$\max_{\mathbf{x}, \tilde{\mathbf{x}} \in \mathcal{C}} \{\|\mathbf{x} - \hat{\mathbf{x}}\|\} \leq D. \tag{9}$$

# 3 Main Result

In this section, we introduce a generic framework to reach an $(\epsilon, \gamma)$-SOSP of the non-convex function $f$ over the convex set $\mathcal{C}$, when $\mathcal{C}$ has a specific structure as we describe below. In particular, we focus on the case when we can solve a quadratic program (QP) of the form

$$\text{minimize} \quad \mathbf{x}^\top \mathbf{A} \mathbf{x} + \mathbf{b}^\top \mathbf{x} + c \qquad \text{subject to} \quad \mathbf{x} \in \mathcal{C}, \tag{10}$$

up to a constant factor $\rho \le 1$ in a finite number of arithmetic operations. Here, $\mathbf{A} \in \mathbb{R}^d$ is a symmetric matrix, $\mathbf{b} \in \mathbb{R}^d$ is a vector, and $c \in \mathbb{R}$ is a scalar. To clarify the notion of solving a problem up to a constant factor $\rho$, consider $\mathbf{x}^*$ as a global minimizer of (10). Then, we say Problem (10) is solved up to a constant factor $\rho \in (0, 1]$ if we have found a feasible solution $\tilde{\mathbf{x}} \in \mathcal{C}$ such that

$$\mathbf{x}^{*\top} \mathbf{A} \mathbf{x}^* + \mathbf{b}^\top \mathbf{x}^* + c \ \le\ \tilde{\mathbf{x}}^\top \mathbf{A} \tilde{\mathbf{x}} + \mathbf{b}^\top \tilde{\mathbf{x}} + c \ \le\ \rho(\mathbf{x}^{*\top} \mathbf{A} \mathbf{x}^* + \mathbf{b}^\top \mathbf{x}^* + c). \tag{11}$$

Note that here w.l.o.g. we have assumed that the optimal objective function value $\mathbf{x}^{*\top} \mathbf{A} \mathbf{x}^* + \mathbf{b}^\top \mathbf{x}^* + c$ is non-positive. Larger constant $\rho$ implies that the approximate solution is more accurate. If $\tilde{\mathbf{x}}$ satisfies the condition in (11), we call it a $\rho$-approximate solution of Problem (10). Indeed, if $\rho = 1$ then $\tilde{\mathbf{x}}$ is a global minimizer of Problem (10).

In Algorithm 1, we introduce a generic framework that achieves an $(\epsilon, \gamma)$-SOSP of Problem (1) whose running time is polynomial in $\epsilon^{-1}$, $\gamma^{-1}$, $\rho^{-1}$ and $d$, when we can find a $\rho$-approximate solution of a quadratic problem of the form (10) in a time that is polynomial in $d$. The proposed scheme consists of two major stages. In the first phase, as mentioned in Steps 2-4, we use a first-order update, i.e., a gradient-based update, to find an $\epsilon$-FOSP, i.e., we update the decision variable $\mathbf{x}$ according to a first-order update until we reach a point $\mathbf{x}_t$ that satisfies the condition

$$\nabla f(\mathbf{x}_t)^\top (\mathbf{x} - \mathbf{x}_t) \ge -\epsilon, \quad \text{for all } \mathbf{x} \in \mathcal{C}. \tag{12}$$

In Section 4, we study in detail projected gradient descent and conditional gradient algorithms for the first order phase of the proposed framework. Interestingly, both of these algorithms require at most $\mathcal{O}(\epsilon^{-2})$ iterations to reach an $\epsilon$-first order stationary point.

The second stage of the proposed scheme uses second-order information of the objective function $f$ to escape from the stationary point if it is a local maximum or a strict saddle point. To be more precise, if we assume that $\mathbf{x}_t$ is a feasible point satisfying the condition (12), we then aim to find a descent direction by solving the following quadratic program

$$\text{minimize} \quad q(\mathbf{u}) := (\mathbf{u} - \mathbf{x}_t)^\top \nabla^2 f(\mathbf{x}_t)(\mathbf{u} - \mathbf{x}_t)$$
$$\text{subject to} \quad \mathbf{u} \in \mathcal{C}, \quad \nabla f(\mathbf{x}_t)^\top (\mathbf{u} - \mathbf{x}_t) = 0, \tag{13}$$

up to a constant factor $\rho$ where $\rho \in (0, 1]$. To be more specific, if we define $q(\mathbf{u}^*)$ as the optimal objective function value of the program in (13), we focus on the cases that we can obtain a feasible point $\mathbf{u}_t$ which is a $\rho$-approximate solution of Problem (13), i.e., $\mathbf{u}_t \in \mathcal{C}$ satisfies the constraints in (13) and

$$q(\mathbf{u}^*) \ \le\ q(\mathbf{u}_t) \ \le\ \rho\, q(\mathbf{u}^*). \tag{14}$$

The problem formulation in (13) can be transformed into the quadratic program in (10); see Section 5 for more details. Note that the constant $\rho$ is independent of $\epsilon$, $\gamma$, and $d$ and only depends on the structure of the convex set $\mathcal{C}$. For instance, if $\mathcal{C}$ is defined in terms of $m$ quadratic constraints one can find a $\rho = m^{-2}$ approximate solution of (13) after at most $\tilde{\mathcal{O}}(md^3)$ arithmetic operations (Section 5).

After computing a feasible point $\mathbf{u}_t$ satisfying the condition in (14), we check the quadratic objective function value at the point $\mathbf{u}_t$, and if the inequality $q(\mathbf{u}_t) < -\rho\gamma$ holds, we follow the update

$$\mathbf{x}_{t+1} = (1 - \sigma)\mathbf{x}_t + \sigma \mathbf{u}_t, \tag{15}$$

where $\sigma$ is a positive stepsize. Otherwise, we stop the process and return $\mathbf{x}_t$ as an $(\epsilon, \gamma)$-second order stationary point of Problem (1). To check this claim, note that Algorithm 1 stops if we reach a point $\mathbf{x}_t$ that satisfies the first-order stationary condition $\nabla f(\mathbf{x}_t)^\top (\mathbf{x} - \mathbf{x}_t) \ge -\epsilon$, and the objective function value for the $\rho$-approximate solution of the quadratic subproblem is larger than $-\rho\gamma$, i.e., $q(\mathbf{u}_t) \ge -\rho\gamma$. The second condition alongside with the fact that $q(\mathbf{u}_t)$ satisfies (14) implies that $q(\mathbf{u}^*) \ge -\gamma$. Therefore, for any $\mathbf{x} \in \mathcal{C}$ and $\nabla f(\mathbf{x}_t)^\top (\mathbf{x} - \mathbf{x}_t) = 0$, it holds that

$$(\mathbf{x} - \mathbf{x}_t)^\top \nabla^2 f(\mathbf{x}_t)(\mathbf{x} - \mathbf{x}_t) \ge -\gamma. \tag{16}$$

---

**Algorithm 1** Generic framework for escaping saddles in constrained optimization

---
**Require:** Stepsize $\sigma > 0$. Initialize $\mathbf{x}_0 \in \mathcal{C}$
1: **for** $t = 1, 2, \ldots$ **do**
2:     **if** $\mathbf{x}_t$ is not an $\epsilon$-first order stationary point **then**
3:         Compute $\mathbf{x}_{t+1}$ using first-order information (Frank-Wolfe or projected gradient descent)
4:     **else**
5:         Find $\mathbf{u}_t$: a $\rho$-approximate solution of (13)
6:         **if** $q(\mathbf{u}_t) < -\rho\gamma$ **then**
7:             Compute the updated variable $\mathbf{x}_{t+1} = (1 - \sigma)\mathbf{x}_t + \sigma\mathbf{u}_t$;
8:         **else**
9:             Return $\mathbf{x}_t$ and stop.
10:         **end if**
11:     **end if**
12: **end for**

---

These two observations show that the outcome of the proposed framework in Algorithm 1 is an $(\epsilon, \gamma)$-SOSP of Problem (1). Now it remains to characterize the number of iterations that Algorithm 1 needs to perform before reaching an $(\epsilon, \gamma)$-SOSP which we formally state in the following theorem.

**Theorem 1.** *Consider the optimization problem defined in* (1). *Suppose that the conditions in Assumptions 1-3 are satisfied. If in the first-order stage, i.e., Steps 2-4, we use the update of Frank-Wolfe or projected gradient descent, the generic framework proposed in Algorithm 1 finds an $(\epsilon, \gamma)$-second-order stationary point of Problem* (1) *after at most $\mathcal{O}(\max\{\epsilon^{-2}, \rho^{-3}\gamma^{-3}\})$ iterations.*

The result in Theorem 1 shows that if the convex constraint $\mathcal{C}$ is such that one can solve the quadratic subproblem in (13) $\rho$-approximately, then the proposed generic framework finds an $(\epsilon, \gamma)$-SOSP point of Problem (1) after at most $\mathcal{O}(\epsilon^{-2})$ first-order and $\mathcal{O}(\rho^{-3}\gamma^{-3})$ second-order updates.

To prove the claim in Theorem 1, we first review first-order conditional gradient and projected gradient algorithms and show that if the current iterate is not a first-order stationary point, by following either of these updates the objective function value decreases by a constant of $\mathcal{O}(\epsilon^2)$ (Section 4). We then focus on the second stage of Algorithm 1 which corresponds to the case that the current iterate is an $\epsilon$-FOSP and we need to solve the quadratic program in (13) approximately (Section 5). In this case, we show that if the iterate is not an $(\epsilon, \gamma)$-SOSP, by following the update in (15) the objective function value decreases at least by a constant of $\mathcal{O}(\rho^3\gamma^3)$. Finally, by combining these two results it can be shown that Algorithm 1 finds an $(\epsilon, \gamma)$-SOSP after at most $\mathcal{O}(\max\{\epsilon^{-2}, \rho^{-3}\gamma^{-3}\})$ iterations.

## 4 First-Order Step: Convergence to a First-Order Stationary Point

In this section, we study two different first-order methods for the first stage of Algorithm 1. The result in this section can also be independently used for convergence to an FOSP of Problem (1) satisfying

$$\nabla f(\mathbf{x}^*)^\top (\mathbf{x} - \mathbf{x}^*) \geq -\epsilon, \qquad \text{for all } \mathbf{x} \in \mathcal{C}, \tag{17}$$

where $\epsilon > 0$ is a positive constant. Although for Algorithm 1 we assume that $\mathcal{C}$ has a specific structure as mentioned in (10), the results in this section hold for any closed and compact convex set $\mathcal{C}$. To keep our result as general as possible, in this section, we study both conditional gradient and projected-based methods when they are used in the first-stage of the proposed generic framework.

### 4.1 Conditional gradient update

The conditional gradient (Frank-Wolfe) update has two steps. We first solve the linear program

$$\mathbf{v}_t = \underset{\mathbf{v} \in \mathcal{C}}{\operatorname{argmax}}\{-\nabla f(\mathbf{x}_t)^\top \mathbf{v}\}. \tag{18}$$

Then, we compute the updated variable $\mathbf{x}_{t+1}$ according to the update

$$\mathbf{x}_{t+1} = (1 - \eta)\mathbf{x}_t + \eta\mathbf{v}_t, \tag{19}$$

where $\eta$ is a stepsize. In the following proposition, we show that if the current iterate is not an $\epsilon$-first order stationary point, then by updating the variable according to (18)-(19) the objective function value decreases. The proof of the following proposition is adopted from [31].

**Proposition 2.** *Consider the optimization problem in* (1). *Suppose Assumptions 1 and 3 hold. Set the stepsize in* (19) *to* $\eta = \epsilon/D^2 L$. *Then, if the iterate* $\mathbf{x}_t$ *at step t is not an $\epsilon$-first order stationary point, the objective function value at the updated variable* $\mathbf{x}_{t+1}$ *satisfies the inequality*

$$f(\mathbf{x}_{t+1}) \leq f(\mathbf{x}_t) - \frac{\epsilon^2}{2D^2 L}. \tag{20}$$

The result in Proposition 2 shows that by following the update of the conditional gradient method the objective function value decreases by $\mathcal{O}(\epsilon^2)$, if an $\epsilon$-FOSP is not achieved.

**Remark 1.** *In step 3 of Algorithm 1 we first check if* $\mathbf{x}_t$ *is an $\epsilon$-FOSP. This can be done by evaluating*

$$\min_{\mathbf{x} \in \mathcal{C}} \{\nabla f(\mathbf{x}_t)^\top (\mathbf{x} - \mathbf{x}_t)\} = \max_{\mathbf{x} \in \mathcal{C}} \{-\nabla f(\mathbf{x}_t)^\top \mathbf{x}\} + \nabla f(\mathbf{x}_t)^\top \mathbf{x}_t \tag{21}$$

*and comparing the optimal value with* $-\epsilon$. *Note that the linear program in* (21) *is the same as the one in* (18). *Therefore, by checking the first-order optimality condition of* $\mathbf{x}_t$, *the variable* $\mathbf{v}_t$ *is already computed, and we need to solve only one linear program per iteration.*

## 4.2 Projected gradient update

The projected gradient descent (PGD) update consists of two steps: (i) descending through the gradient direction and (ii) projecting the updated variable onto the convex constraint set. These two steps can be combined together and the update can be explicitly written as

$$\mathbf{x}_{t+1} = \pi_{\mathcal{C}} \{\mathbf{x}_t - \eta \nabla f(\mathbf{x}_t)\}, \tag{22}$$

where $\pi_{\mathcal{C}}(.)$ is the Euclidean projection onto the convex set $\mathcal{C}$ and $\eta$ is a positive stepsize. In the following proposition, we first show that by following the update of PGD the objective function value decreases by a constant until we reach an $\epsilon$- FOSP. Further, we show that the number of required iterations for PGD to reach an $\epsilon$-FOSP is of $\mathcal{O}(\epsilon^{-2})$.

**Proposition 3.** *Consider Problem* (1). *Suppose Assumptions 1 and 3 are satisfied. Further, assume that the gradients* $\nabla f(\mathbf{x})$ *are uniformly bounded by K for all* $\mathbf{x} \in \mathcal{C}$. *If the stepsize of the projected gradient descent method defined in* (22) *is set to* $\eta = 1/L$ *the objective function value decreases by*

$$f(\mathbf{x}_{t+1}) \leq f(\mathbf{x}_t) - \frac{\epsilon^2 L}{2(K + LD)^2}, \tag{23}$$

*Moreover, iterates reach a first-order stationary point satisfying* (17) *after at most* $\mathcal{O}(\epsilon^{-2})$ *iterations.*

Proposition 3 shows that by following the update of PGD the function value decreases by $\mathcal{O}(\epsilon^2)$ until we reach an $\epsilon$-FOSP. It further shows PGD obtains an $\epsilon$-FOSP satisfying (17) after at most $\mathcal{O}(\epsilon^{-2})$ iterations. To the best of our knowledge, this result is also novel, since the only convergence guarantee for PGD in [26] is in terms of number of iterations to reach a point with a gradient mapping norm less than $\epsilon$, while our result characterizes number of iterations to satisfy (17).

**Remark 2.** *To use the PGD update in the first stage of Algorithm 1 one needs to define a criteria to check if* $\mathbf{x}_t$ *is an $\epsilon$-FOSP or not. However, in PGD we do not solve the linear program* $\min_{\mathbf{x} \in \mathcal{C}} \{\nabla f(\mathbf{x}_t)^\top (\mathbf{x} - \mathbf{x}_t)\}$. *This issue can be resolved by checking the condition* $\|\mathbf{x}_t - \mathbf{x}_{t+1}\| \leq \epsilon/(K + LD)$ *which is a sufficient condition for the condition in* (17). *In other words, if this condition holds we stop and* $\mathbf{x}_t$ *is an $\epsilon$-FOSP; otherwise, the result in* (23) *holds and the function value decreases. For more details please check the proof of Proposition 3.*

## 5 Second-Order Step: Escape from Saddle Points

In this section, we study the second stage of the framework in Algorithm 1 which corresponds to the case that the current iterate is an $\epsilon$-FOSP. Note that when we reach a critical point the goal is to find a feasible point $\mathbf{u} \in \mathcal{C}$ in the tangent space $\nabla f(\mathbf{x}_t)^\top (\mathbf{u} - \mathbf{x}_t) = 0$ that makes the inner product $(\mathbf{u} - \mathbf{x}_t)^\top \nabla^2 f(\mathbf{x}_t)(\mathbf{u} - \mathbf{x}_t)$ smaller than $-\gamma$. To achieve this goal we need to check the minimum value of this inner product over the constraints, i.e., we need to solve the quadratic program in (13) up to a constant factor $\rho \in (0, 1]$. In the following proposition, we show that the updated variable according to (15) decreases the objective function value if the condition $q(\mathbf{u}_t) < -\rho\gamma$ holds.

**Proposition 4.** *Consider the quadratic program in* (13). *Let* $\mathbf{u}_t$ *be a $\rho$-approximate solution for quadratic subproblem in* (13). *Suppose that Assumptions 2 and 3 hold. Further, set the stepsize $\sigma = \rho\gamma/MD^3$. If the quadratic objective function value $q$ evaluated at $\mathbf{u}_t$ satisfies the condition $q(\mathbf{u}_t) < -\rho\gamma$, then the updated variable according to* (15) *satisfies the inequality*

$$f(\mathbf{x}_{t+1}) \le f(\mathbf{x}_t) - \frac{\rho^3\gamma^3}{3M^2D^6}. \tag{24}$$

The only unanswered question is how to solve the quadratic subproblem in (13) up to a constant factor $\rho \in (0,1]$. For general $\mathcal{C}$, the quadratic subproblem could be NP-hard [34]; however, for some special choices of the convex constraint $\mathcal{C}$, this quadratic program (QP) can be solved either exactly or approximately up to a constant factor. In the following section, we focus on the quadratic constraint case, but indeed there are other classes of constraints that satisfy our required condition.

### 5.1 Quadratic constraints case

In this section, we focus on the case where the constraint set $\mathcal{C}$ is defined as the intersection of $m$ ellipsoids centered at the origin.[1] In particular, assume that the set $\mathcal{C}$ is given by

$$\mathcal{C} := \{\mathbf{x} \in \mathbb{R}^d \mid \mathbf{x}^\top \mathbf{Q}_i \mathbf{x} \le 1, \text{ for all } i = 1,\dots,m\}, \tag{25}$$

where $\mathbf{Q}_i \in \mathbb{S}_+^d$. Under this assumption, the QP in (13) can be written as

$$\min_{\mathbf{u}} \quad (\mathbf{u} - \mathbf{x}_t)^\top \nabla^2 f(\mathbf{x}_t)(\mathbf{u} - \mathbf{x}_t)$$
$$\text{s.t.} \quad \mathbf{u}^\top \mathbf{Q}_i \mathbf{u} \le 1, \quad \text{for } i = 1,\dots,m \quad \text{and} \quad \nabla f(\mathbf{x}_t)^\top (\mathbf{u} - \mathbf{x}_t) = 0. \tag{26}$$

Note that the equality constraint $\nabla f(\mathbf{x}_t)^\top (\mathbf{u} - \mathbf{x}_t) = 0$ does not change the hardness of the problem and can be easily eliminated. To do so, first define a new optimization variable $\mathbf{z} := \mathbf{u} - \mathbf{x}_t$ to obtain

$$\min_{\mathbf{z}} \quad \mathbf{z}^\top \nabla^2 f(\mathbf{x}_t)\mathbf{z}$$
$$\text{s.t.} \quad (\mathbf{z} + \mathbf{x}_t)^\top \mathbf{Q}_i(\mathbf{z} + \mathbf{x}_t) \le 1, \quad \text{for } i = 1,\dots,m \quad \text{and} \quad \nabla f(\mathbf{x}_t)^\top \mathbf{z} = 0, \tag{27}$$

Then, find a basis for the tangent space $\nabla f(\mathbf{x}_t)^\top \mathbf{z} = 0$. Indeed, using the Gramm-Schmidt procedure, we can find an orthonormal basis for the space $\mathbb{R}^d$ of the form $\{\mathbf{v}_1, \dots, \mathbf{v}_{d-1}, \frac{\nabla f(\mathbf{x}_t)}{\|\nabla f(\mathbf{x}_t)\|}\}$ at the complexity of $\mathcal{O}(d^3)$. If we define $\mathbf{A} = [\mathbf{v}_1; \dots; \mathbf{v}_{d-1}] \in \mathbb{R}^{d \times d-1}$ as the concatenation of the vectors $\{\mathbf{v}_1, \dots, \mathbf{v}_{d-1}\}$, then any vector $\mathbf{z}$ satisfying $\nabla f(\mathbf{x}_t)^\top \mathbf{z} = 0$ can be written as $\mathbf{z} = \mathbf{A}\mathbf{y}$ where $\mathbf{y} \in \mathbb{R}^{d-1}$. Hence, (27) is equivalent to

$$\min_{\mathbf{z}} \quad \mathbf{y}^\top \mathbf{A}^\top \nabla^2 f(\mathbf{x}_t)\mathbf{A}\mathbf{y} \quad \text{s.t.} \quad (\mathbf{A}\mathbf{y} + \mathbf{x}_t)^\top \mathbf{Q}_i(\mathbf{A}\mathbf{y} + \mathbf{x}_t) \le 1, \quad \text{for } i = 1,\dots,m. \tag{28}$$

This procedure reduces the dimension of the problem from $d$ to $d-1$. It is not hard to check that the center of ellipsoids in (28) is $-\mathbf{A}^\top \mathbf{x}_t$. By a simple change of variable $\mathbf{A}\hat{\mathbf{y}} := \mathbf{A}\mathbf{y} + \mathbf{x}_t$ we obtain

$$\min_{\mathbf{z}} \quad \hat{\mathbf{y}}^\top \mathbf{A}^\top \nabla^2 f(\mathbf{x}_t)\mathbf{A}\hat{\mathbf{y}} - 2\mathbf{x}_t^\top \nabla^2 f(\mathbf{x}_t)\mathbf{A}\hat{\mathbf{y}} + \mathbf{x}_t^\top \nabla^2 f(\mathbf{x}_t)\mathbf{x}_t$$
$$\text{s.t.} \quad \hat{\mathbf{y}}^\top \mathbf{A}^\top \mathbf{Q}_i\mathbf{A}\hat{\mathbf{y}} \le 1, \quad \text{for } i = 1,\dots,m. \tag{29}$$

Define the matrices $\tilde{\mathbf{Q}}_i := \mathbf{A}^\top \mathbf{Q}_i \mathbf{A}$ and $\mathbf{B}_t := \mathbf{A}^\top \nabla^2 f(\mathbf{x}_t)\mathbf{A}$, the vector $\mathbf{s}_t = -2\mathbf{x}_t^\top \nabla^2 f(\mathbf{x}_t)\mathbf{A}$, and the scalar $\mathbf{c}_t := \mathbf{x}_t^\top \nabla^2 f(\mathbf{x}_t)\mathbf{x}_t$. Using these definitions the problem reduces to

$$\min_{\mathbf{z}} \quad q(\hat{\mathbf{y}}) := \hat{\mathbf{y}}^\top \mathbf{B}_t\hat{\mathbf{y}} + \mathbf{s}_t^\top \hat{\mathbf{y}} + c_t \qquad \text{s.t.} \quad \hat{\mathbf{y}}^\top \tilde{\mathbf{Q}}_i\hat{\mathbf{y}} \le 1, \quad \text{for } i = 1,\dots,m. \tag{30}$$

Note that the matrices $\tilde{\mathbf{Q}}_i \in \mathbb{S}_+^d$ are positive semidefinite, while the matrix $\mathbf{B}_t \in \mathbb{S}^d$ might be indefinite. Indeed, the optimal objective function value of the program in (30) is equal to the optimal objective function value of (26). Further, note that if we find a $\rho$-approximate solution $\hat{\mathbf{y}}^*$ for (30), we can recover a $\rho$-approximate solution $\mathbf{u}^*$ for (26) using the transformation $\mathbf{u}^* = \mathbf{A}\hat{\mathbf{y}}^*$.

The program in (30) is a specific *Quadratic Constraint Quadratic Program* (QCQP), where all the constraints are centered at $\mathbf{0}$. For the specific case of $m = 1$, the duality gap of this problem is zero and simply by transferring the problem to the dual domain one can solve Problem (30) exactly. In the following proposition, we focus on the general case of $m \geq 1$ and explain how to find a $\rho$-approximate solution for (30).

**Proposition 5.** *Consider Problem* (30) *and define* $q_{min}$ *as the minimum objective value of the problem. Based on the result in [22], there exists a polynomial time method that obtains a point* $\hat{\mathbf{y}}^*$

$$q(\hat{\mathbf{y}}^*) \leq \frac{1-\zeta}{m^2(1+\zeta)^2} \, q_{min} + \left(1 - \frac{1-\zeta}{m^2(1+\zeta)^2}\right) \mathbf{x}_t^\top \nabla^2 f(\mathbf{x}_t)\mathbf{x}_t \qquad (31)$$

*after at most* $\mathcal{O}(d^3(m\log(1/\delta) + \log(1/\zeta) + \log d))$ *arithmetic operations, where* $\delta$ *is the ratio of the radius of the largest inscribing sphere over that of the smallest circumscribing sphere of the feasible set. Further, based on [44], using a SDP relaxation of* (30) *one can find a point* $\hat{\mathbf{y}}^*$ *such that*

$$q(\hat{\mathbf{y}}^*) \leq \frac{1}{m} \, q_{min} + \left(1 - \frac{1}{m}\right) \mathbf{x}_t^\top \nabla^2 f(\mathbf{x}_t)\mathbf{x}_t. \qquad (32)$$

*Proof.* If we define the function $\tilde{q}$ as $\tilde{q}(\mathbf{x}) := q(\mathbf{x}) - c_t$, using the approaches in [22] and [44], we can find a $\rho$ approximate solution for $\min_{\hat{\mathbf{y}}} \tilde{q}(\hat{\mathbf{y}})$ subject to $\hat{\mathbf{y}}^\top \tilde{\mathbf{Q}}_i \hat{\mathbf{y}} \leq 1$ for $i = 1, \ldots, m$. In other words, we can find a point $\hat{\mathbf{y}}^*$ such that $\tilde{q}(\hat{\mathbf{y}}^*) \leq \rho \, \tilde{q}_{min}$ where $0 < \rho < 1$ and $\tilde{q}_{min}$ is the minimum objective function value of $\tilde{q}$ over the constraint set which satisfies $\tilde{q}_{min} = q_{min} - c_t$. Replacing $\tilde{q}(\hat{\mathbf{y}}^*)$ and $\tilde{q}_{min}$ by their definitions and regrouping the terms imply that $\hat{\mathbf{y}}^*$ satisfies the condition $q(\hat{\mathbf{y}}^*) \leq \rho q_{min} + (1 - \rho)c_t$. Replacing $\rho$ by $\frac{1-\zeta}{m^2(1+\zeta)^2}$ (which is the constant factor approximation shown in [22]) leads to the claim in (31), and substituting $\rho$ by $1/m$ (which is the approximation bound in [44]) implies the result in (32). $\qquad \square$

The result in Proposition 5 indicates that if $\mathbf{x}_t^\top \nabla^2 f(\mathbf{x}_t)\mathbf{x}_t$ is non-positive, then one can find a $\rho$-approximate solution for Problem (30) and consequently Problem (26). This condition is satisfied if we assume that $\max_{\mathbf{x} \in \mathcal{C}} \mathbf{x}^\top \nabla^2 f(\mathbf{x})\mathbf{x} \leq 0$. For instance, for a concave minimization problem over the convex set $\mathcal{C}$ this condition is satisfied. In fact, it can be shown that our analysis still stands even if $\max_{\mathbf{x} \in \mathcal{C}} \mathbf{x}^\top \nabla^2 f(\mathbf{x})\mathbf{x}$ is at most $\mathcal{O}(\gamma)$. Note that this condition is significantly weaker than requiring the function to be concave when restricted to the feasible set. The condition essentially implies that the quadratic term in the Taylor expansion of the function evaluated at the origin should be negative (or not too positive).

**Corollary 1.** *Consider a convex set* $\mathcal{C}$ *which is defined as the intersection of* $m \geq 1$ *ellipsoids centered at the origin. Further, assume that the objective function Hessian* $\nabla^2 f$ *satisfies the condition* $\max_{\mathbf{x} \in \mathcal{C}} \mathbf{x}^\top \nabla^2 f(\mathbf{x})\mathbf{x} \leq 0$. *Then, for* $\rho = 1/m$ *and* $\rho = 1/m^2$, *it is possible to find a* $\rho$-approximate *solution of Problem* (13) *in time polynomial in* $m$ *and* $d$.

By using the approach in [22], we can solve the QCQP in (29) with the approximation factor $\rho \approx 1/m^2$ for $m \geq 1$ at the overall complexity of $\tilde{\mathcal{O}}(md^3)$ when the constraint $\mathcal{C}$ is defined as $m$ convex quadratic constraints. As the total number of calls to the second-order stage is at most $\mathcal{O}(\rho^{-3}\gamma^{-3}) = \mathcal{O}(m^6\gamma^{-3})$, we obtain that the total number of arithmetic operations for the second-order stage is at most $\tilde{\mathcal{O}}(m^7d^3\gamma^{-3})$. The constant factor can be improved to $1/m$ if we solve the SDP relaxation problem suggested in [44].

## 6 Stochastic Extension

In this section, we focus on stochastic constrained minimization problems. Consider the optimization problem in (1) when the objective function $f$ is defined as an expectation of a set of stochastic functions $F : \mathbb{R}^d \times \mathbb{R}^r \to \mathbb{R}$ with inputs $\mathbf{x} \in \mathbb{R}^d$ and $\mathbf{\Theta} \in \mathbb{R}^r$, where $\mathbf{\Theta}$ is a random variable with probability distribution $\mathcal{P}$. To be more precise, we consider the optimization problem

$$\text{minimize } f(\mathbf{x}) := \mathbb{E}\left[F(\mathbf{x}, \mathbf{\Theta})\right], \qquad \text{subject to } \mathbf{x} \in \mathcal{C}. \qquad (33)$$

Our goal is to find a point which satisfies the necessary optimality conditions with high probability.

## Algorithm 2

**Require:** Stepsize $\sigma_t > 0$. Initialize $\mathbf{x}_0 \in \mathcal{C}$
1: **for** $t = 1, 2, \ldots$ **do**
2:     Compute $\mathbf{v}_t = \arg\max_{\mathbf{v} \in \mathcal{C}} \{-\mathbf{d}_t^\top \mathbf{v}\}$
3:     **if** $\mathbf{d}_t^\top (\mathbf{v}_t - \mathbf{x}_t) < -\epsilon/2$ **then**
4:         Compute $\mathbf{x}_{t+1} = (1 - \eta)\mathbf{x}_t + \eta \mathbf{v}_t$
5:     **else**
6:         Find $\mathbf{u}_t$: a $\rho$-approximate solution of
            $\min_{\mathbf{u}} \ (\mathbf{u} - \mathbf{x}_t)^\top \mathbf{H}_t (\mathbf{u} - \mathbf{x}_t)$      s.t. $\mathbf{u} \in \mathcal{C}, \ \mathbf{d}_t^\top (\mathbf{u} - \mathbf{x}_t) \leq r.$
7:         **if** $q(\mathbf{u}_t) < -\rho\gamma/2$ **then**
8:             Compute the updated variable $\mathbf{x}_{t+1} = (1 - \sigma)\mathbf{x}_t + \sigma \mathbf{u}_t$;
9:         **else**
10:        Return $\mathbf{x}_t$ and stop.
11:       **end if**
12:    **end if**
13: **end for**

Consider the vector $\mathbf{d}_t = (1/b_g) \sum_{i=1}^{b_g} \nabla F(\mathbf{x}_t, \boldsymbol{\theta}_i)$ and matrix $\mathbf{H}_t = (1/b_H) \sum_{i=1}^{b_H} \nabla^2 F(\mathbf{x}_t, \boldsymbol{\theta}_i)$ as stochastic approximations of the gradient $\nabla f(\mathbf{x}_t)$ and Hessian $\nabla^2 f(\mathbf{x}_t)$, respectively. Here $b_g$ and $b_H$ are the gradient and Hessian batch sizes, respectively, and the vectors $\boldsymbol{\theta}_i$ are the realizations of the random variable $\boldsymbol{\Theta}$. In Algorithm 2, we present the stochastic variant of our proposed scheme for finding an $(\epsilon, \gamma)$-SOSP of Problem (33). Algorithm 2 differs from Algorithm 1 in using the stochastic gradients $\mathbf{d}_t$ and Hessians $\mathbf{H}_t$ in lieu of the exact gradients $\nabla f(\mathbf{x}_t)$ and $\nabla^2 f(\mathbf{x}_t)$ Hessians. The second major difference is the inequality constraint in step 6. Here instead of using the constraint $\mathbf{d}_t^\top (\mathbf{u} - \mathbf{x}_t) = 0$ we need to use $\mathbf{d}_t^\top (\mathbf{u} - \mathbf{x}_t) \leq r$, where $r > 0$ is a properly chosen constant. This modification is needed to ensure that if a point satisfies this constraint with high probability it also satisfies the constraint $\nabla f(\mathbf{x}_t)^\top (\mathbf{u} - \mathbf{x}_t) = 0$. This modification implies that we need to handle a linear inequality constraint instead of the linear equality constraint, which is computationally manageable for some constraints including the case that $\mathcal{C}$ is a single ball constraint [28]. To prove our main result we assume that the following conditions also hold.

**Assumption 4.** *The variance of the stochastic gradients and Hessians are uniformly bounded by constants $\nu^2$ and $\xi^2$, respectively, i.e., for any $\mathbf{x} \in \mathcal{C}$ and $\boldsymbol{\theta}$ we can write*

$$\mathbb{E}\left[\|\nabla F(\mathbf{x}, \boldsymbol{\theta}) - \nabla f(\mathbf{x})\|^2\right] \leq \nu^2, \qquad \mathbb{E}\left[\|\nabla^2 F(\mathbf{x}, \boldsymbol{\theta}) - \nabla^2 f(\mathbf{x})\|^2\right] \leq \xi^2. \qquad (34)$$

The required conditions in Assumption 4 ensure that the variances of stochastic gradients and Hessians are uniformly bounded above, which are customary in stochastic optimizaiton.

In the following theorem, we characterize the iteration complexity of Algorithm 2 to reach an $(\epsilon, \gamma)$-SOSP of Problem (33) with high probability.

**Theorem 2.** *Consider the optimization problem in* (33)*. Suppose the conditions in Assumptions 1-4 are satisfied. If the batch sizes are $b_g = \mathcal{O}(\max\{\rho^{-4}\gamma^{-4}, \epsilon^{-2}\})$ and $b_H = \mathcal{O}(\rho^{-2}\gamma^{-2})$ and we set the parameter $r = \mathcal{O}(\rho^2\gamma^2)$, then the outcome of the proposed framework outlined in Algorithm 2 is an $(\epsilon, \gamma)$-second-order stationary point of Problem* (33) *with high probability. Further, the total number of iterations to reach such point is at most $\mathcal{O}(\max\{\epsilon^{-2}, \rho^{-3}\gamma^{-3}\})$ with high probability.*

The result in Theorem 2 indicates that the total number of iterations to reach an $(\epsilon, \gamma)$-SOSP is at most $\mathcal{O}(\max\{\epsilon^{-2}, \rho^{-3}\gamma^{-3}\})$. As each iteration at most requires $\mathcal{O}(\max\{\rho^{-4}\gamma^{-4}, \epsilon^{-2}\})$ stochastic gradient and $\mathcal{O}(\rho^{-2}\gamma^{-2})$ stochastic Hessian evaluations, the total number of stochastic gradient and Hessian computations to reach an $(\epsilon, \gamma)$-SOSP is of $\mathcal{O}(\max\{\epsilon^{-2}\rho^{-4}\gamma^{-4}, \epsilon^{-4}, \rho^{-7}\gamma^{-7}\})$ and $\mathcal{O}(\max\{\epsilon^{-2}\rho^{-3}\gamma^{-3}, \rho^{-5}\gamma^{-5}\})$, respectively.

## Acknowledgment

This work was supported by DARPA Lagrange and ONR BRC Program. The authors would like to thank Yue Sun for pointing out a missing condition in the first draft of the paper.

## Footnotes

[1]To simplify the constant factor approximation $\rho$ we assume ellipsoids are centered at the origin. If we drop this assumption then $\rho$ will depend on the maximum distance between the origin and the boundary of each of the ellipsoids, e.g., see equation (6) in [44].

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
