[Supplementary Material · final_supp_mat_NeurIPS.pdf]

# 7 Supplementary Material

## 7.1 Proof of Proposition 1

The claim in (2) follows from Proposition 2.1.2 in [4]. The proof for the claim in (3) is similar to the proof of Proposition 2.1.2 in [4], and we mention it for completeness.

We prove the claim in (3) by contradiction. Suppose that $(\mathbf{x} - \mathbf{x}^*)^\top \nabla^2 f(\mathbf{x}^*)(\mathbf{x} - \mathbf{x}^*) < 0$ for some $\mathbf{x} \in \mathcal{C}$ satisfying $\nabla f(\mathbf{x}^*)^\top (\mathbf{x} - \mathbf{x}^*) = 0$. By the mean value theorem, for any $\epsilon > 0$ there exists an $\alpha \in [0, 1]$ such that

$$f(\mathbf{x}^* + \epsilon(\mathbf{x} - \mathbf{x}^*))$$
$$= f(\mathbf{x}^*) + \epsilon \nabla f(\mathbf{x}^*)^\top (\mathbf{x} - \mathbf{x}^*) + \epsilon^2 (\mathbf{x} - \mathbf{x}^*) \nabla^2 f(\mathbf{x}^* + \alpha \epsilon(\mathbf{x} - \mathbf{x}^*))^\top (\mathbf{x} - \mathbf{x}^*), \quad (35)$$

Use the relation $\nabla f(\mathbf{x}^*)^\top (\mathbf{x} - \mathbf{x}^*) = 0$ to simplify the right hand side to

$$f(\mathbf{x}^* + \epsilon(\mathbf{x} - \mathbf{x}^*)) = f(\mathbf{x}^*) + \epsilon^2 (\mathbf{x} - \mathbf{x}^*) \nabla^2 f(\mathbf{x}^* + \alpha \epsilon(\mathbf{x} - \mathbf{x}^*))^\top (\mathbf{x} - \mathbf{x}^*). \quad (36)$$

Note that since $(\mathbf{x} - \mathbf{x}^*)^\top \nabla^2 f(\mathbf{x}^*)(\mathbf{x} - \mathbf{x}^*) < 0$ and the Hessian is continuous, we have for all sufficiently small $\epsilon > 0$, $(\mathbf{x} - \mathbf{x}^*) \nabla^2 f(\mathbf{x}^* + \alpha \epsilon(\mathbf{x} - \mathbf{x}^*))^\top (\mathbf{x} - \mathbf{x}^*) < 0$. This observation and the expression in (36) follows that for sufficiently small $\epsilon$ we have $f(\mathbf{x}^* + \epsilon(\mathbf{x} - \mathbf{x}^*)) < f(\mathbf{x}^*)$. Note that the point $\mathbf{x}^* + \epsilon(\mathbf{x} - \mathbf{x}^*)$ for all $\epsilon \in [0, 1]$ belongs to the set $\mathcal{C}$ and satisfies the inequality $\nabla f(\mathbf{x}^*)^\top ((\mathbf{x}^* + \epsilon(\mathbf{x} - \mathbf{x}^*)) - \mathbf{x}^*) = 0$. Therefore, we obtained a contradiction of the local optimality of $\mathbf{x}^*$.

## 7.2 Proof of Proposition 2

First consider the definition $G(\mathbf{x}_t) = \max_{\mathbf{x} \in \mathcal{C}} \{ -\nabla f(\mathbf{x}_t)^\top (\mathbf{x} - \mathbf{x}_t) \}$ which is also known as Frank-Wolfe gap [31]. This constant measures how close the point $\mathbf{x}_t$ is to be a first-order stationary point. If $G(\mathbf{x}_t) \leq \epsilon$, then $\mathbf{x}_t$ is an $\epsilon$-first-order stationary point. Let's assume that $G(\mathbf{x}_t) > \epsilon$. Then, based on the Lipschitz continuity of gradients and the definition of $G(\mathbf{x}_t)$ we can write

$$f(\mathbf{x}_{t+1}) \leq f(\mathbf{x}_t) + \nabla f(\mathbf{x}_t)^\top (\mathbf{x}_{t+1} - \mathbf{x}_t) + \frac{L}{2} \|\mathbf{x}_{t+1} - \mathbf{x}_t\|^2$$
$$= f(\mathbf{x}_t) + \eta \nabla f(\mathbf{x}_t)^\top (\mathbf{v}_t - \mathbf{x}_t) + \frac{L\eta^2}{2} \|\mathbf{v}_t - \mathbf{x}_t\|^2$$
$$\leq f(\mathbf{x}_t) - \eta G(\mathbf{x}_t) + \frac{\eta^2 D^2 L}{2}, \quad (37)$$

where the last inequality follows from $\|\mathbf{v}_t - \mathbf{x}_t\| \leq D$. Replacing the stepsize $\eta$ by its value $\epsilon/D^2 L$ and $G(\mathbf{x}_t)$ by its lower bound $\epsilon$ lead to

$$f(\mathbf{x}_{t+1}) \leq f(\mathbf{x}_t) - \frac{\epsilon^2}{2D^2 L}. \quad (38)$$

This result implies that if the current point $\mathbf{x}_t$ is not an $\epsilon$-first order stationary point, by following the update of Frank-Wolfe algorithm the objective function value decreases by $\epsilon^2/2D^2 L$. Therefore, after at most $2D^2 L(f(\mathbf{x}_0) - f(\mathbf{x}^*))/\epsilon^2$ iterations we either reach the global minimum or one of the iterates $\mathbf{x}_t$ satisfies $G(\mathbf{x}_t) \leq \epsilon$ which implies that

$$\nabla f(\mathbf{x}_t)^\top (\mathbf{x} - \mathbf{x}_t) \geq -\epsilon, \qquad \text{for all } \mathbf{x} \in \mathcal{C}, \quad (39)$$

and the claim in Proposition 2 follows.

## 7.3 Proof of Proposition 3

First note, that based on the projection property we know that

$$(\mathbf{x}_t - \eta \nabla f(\mathbf{x}_t) - \mathbf{x}_{t+1})^\top (\mathbf{x} - \mathbf{x}_{t+1}) \leq 0, \qquad \text{for all } \mathbf{x} \in \mathcal{C}. \quad (40)$$

Therefore, by setting $\mathbf{x} = \mathbf{x}_t$ we obtain that

$$\eta \nabla f(\mathbf{x}_t)^\top (\mathbf{x}_{t+1} - \mathbf{x}_t) \leq -\|\mathbf{x}_t - \mathbf{x}_{t+1}\|^2. \quad (41)$$

Hence, we can replace the inner product $\nabla f(\mathbf{x}_t)^\top (\mathbf{x}_{t+1} - \mathbf{x}_t)$ by its upper bound $-\|\mathbf{x}_t - \mathbf{x}_{t+1}\|^2/\eta$

$$f(\mathbf{x}_{t+1}) \le f(\mathbf{x}_t) + \nabla f(\mathbf{x}_t)^\top (\mathbf{x}_{t+1} - \mathbf{x}_t) + \frac{L}{2}\|\mathbf{x}_{t+1} - \mathbf{x}_t\|^2$$

$$\le f(\mathbf{x}_t) - \frac{\|\mathbf{x}_t - \mathbf{x}_{t+1}\|^2}{\eta} + \frac{L}{2}\|\mathbf{x}_{t+1} - \mathbf{x}_t\|^2$$

$$= f(\mathbf{x}_t) - \frac{L}{2}\|\mathbf{x}_{t+1} - \mathbf{x}_t\|^2, \tag{42}$$

where the equality follows by setting $\eta = 1/L$. Indeed, if $\mathbf{x}_{t+1} = \mathbf{x}_t$ then we are at a first-order stationary point, however, we need a finite time analysis. To do so, note that for any $\mathbf{x} \in \mathcal{C}$ we have

$$(\mathbf{x}_t - \eta \nabla f(\mathbf{x}_t) - \mathbf{x}_{t+1})^\top (\mathbf{x} - \mathbf{x}_{t+1}) \le 0. \tag{43}$$

Therefore, for any $\mathbf{x} \in \mathcal{C}$ it holds

$$\nabla f(\mathbf{x}_t)^\top (\mathbf{x} - \mathbf{x}_{t+1}) \ge L(\mathbf{x}_t - \mathbf{x}_{t+1})^\top (\mathbf{x} - \mathbf{x}_{t+1}), \tag{44}$$

which implies that

$$\nabla f(\mathbf{x}_t)^\top (\mathbf{x} - \mathbf{x}_t) \ge \nabla f(\mathbf{x}_t)^\top (\mathbf{x}_{t+1} - \mathbf{x}_t) + L(\mathbf{x}_t - \mathbf{x}_{t+1})^\top (\mathbf{x} - \mathbf{x}_{t+1})$$

$$\ge -K\|\mathbf{x}_{t+1} - \mathbf{x}_t\| - LD\|\mathbf{x}_t - \mathbf{x}_{t+1}\|$$

$$\ge -(K + LD)\|\mathbf{x}_t - \mathbf{x}_{t+1}\|, \tag{45}$$

where $K$ is an upper bound on the norm of gradient over the convex set $\mathcal{C}$. Therefore, we can write

$$\min_{\mathbf{x} \in \mathcal{C}} \nabla f(\mathbf{x}_t)^\top (\mathbf{x} - \mathbf{x}_t) \ge -(K + LD)\|\mathbf{x}_t - \mathbf{x}_{t+1}\|, \tag{46}$$

Combining these results, we obtain that we should check the norm $\|\mathbf{x}_t - \mathbf{x}_{t+1}\|$ at each iteration and check whether if it is larger than $\epsilon/(K + LD)$ or not. If the norm is larger than the threshold then

$$f(\mathbf{x}_{t+1}) \le f(\mathbf{x}_t) - \frac{\epsilon^2 L}{2(K + LD)^2}. \tag{47}$$

If the norm is smaller than the threshold then we stop and the iterate $\mathbf{x}_t$ satisfies the inequality

$$\nabla f(\mathbf{x}_t)^\top (\mathbf{x} - \mathbf{x}_t) \ge -\epsilon, \qquad \text{for all } \mathbf{x} \in \mathcal{C}. \tag{48}$$

Note that this process can not take more than $\mathcal{O}(\frac{f(\mathbf{x}_0) - f(\mathbf{x}^*)}{\epsilon^2})$ iterations.

## 7.4 Proof of Proposition 4

The Taylor's expansion of the function $f$ around the point $\mathbf{x}_t$ and $M$-Lipschitz continuity of the Hessians imply that

$$f(\mathbf{x}_{t+1}) \le f(\mathbf{x}_t) + \nabla f(\mathbf{x}_t)^\top (\mathbf{x}_{t+1} - \mathbf{x}_t) + \frac{1}{2}(\mathbf{x}_{t+1} - \mathbf{x}_t)^\top \nabla^2 f(\mathbf{x})(\mathbf{x}_{t+1} - \mathbf{x}_t) + \frac{M}{6}\|\mathbf{x}_{t+1} - \mathbf{x}_t\|^3. \tag{49}$$

Replace $\mathbf{x}_{t+1} - \mathbf{x}_t$ by the expression $\sigma(\mathbf{u}_t - \mathbf{x}_t)$ to obtain

$$f(\mathbf{x}_{t+1}) \le f(\mathbf{x}_t) + \sigma \nabla f(\mathbf{x}_t)^\top (\mathbf{u}_t - \mathbf{x}_t) + \frac{\sigma^2}{2}(\mathbf{u}_t - \mathbf{x}_t)^\top \nabla^2 f(\mathbf{x})(\mathbf{u}_t - \mathbf{x}_t) + \frac{M\sigma^3}{6}\|\mathbf{u}_t - \mathbf{x}_t\|^3. \tag{50}$$

Since, $\mathbf{u}_t$ is a $\rho$-approximate solution for the subproblem in (13) with the objective function value $q(\mathbf{u}_t) \le -\rho\gamma$, we can substitute the quadratic term $(\mathbf{u}_t - \mathbf{x}_t)^\top \nabla^2 f(\mathbf{x})(\mathbf{u}_t - \mathbf{x}_t)$ by its upper bound $-\rho\gamma$. Additionally, the vector $\mathbf{u}_t$ is chosen such that $\nabla f(\mathbf{x}_t)^\top (\mathbf{u}_t - \mathbf{x}_t) = 0$ and therefore the linear term in (50) can be eliminated. Further, the cubic term $\|\mathbf{u}_t - \mathbf{x}_t\|^3$ is upper bounded by $D^3$ since both $\mathbf{u}_t$ and $\mathbf{x}_t$ belong to the convex set $\mathcal{C}$. Applying these substitutions into (50) yields

$$f(\mathbf{x}_{t+1}) \le f(\mathbf{x}_t) - \frac{\sigma^2 \rho\gamma}{2} + \frac{\sigma^3 M D^3}{6}. \tag{51}$$

By setting $\sigma = \rho\gamma/MD^3$ in (51) it follows that

$$f(\mathbf{x}_{t+1}) \le f(\mathbf{x}_t) - \frac{\rho^3\gamma^3}{2M^2 D^6} + \frac{\rho^3\gamma^3}{6M^2 D^6}$$

$$= f(\mathbf{x}_t) - \frac{\rho^3\gamma^3}{3M^2 D^6}. \tag{52}$$

Therefore, in this case, the objective function value decreases at least by a fixed value of $\mathcal{O}(\rho^3\gamma^3)$.

## 7.5 Proof of Theorem 1

Then at each iteration, either the first oder optimality condition is not satisfied and the function value decreases by a constant of $\mathcal{O}(\epsilon^2)$, or this condition is satisfied and we use a second-order update which leads to a objective function value decrement of $\mathcal{O}(\rho^3\gamma^3)$. This shows that if have not reached an $(\epsilon,\gamma)$-second order stationary point the objective function value decreases at least by $\mathcal{O}(\min\{\epsilon^2,\rho^3\gamma^3\})$. Therefore, we either reach the global minimum or converge to an $(\epsilon,\gamma)$-second order stationary point of Problem (1) after at most $\mathcal{O}\left(\frac{f(\mathbf{x}_0)-f(\mathbf{x}^*)}{\min\{\epsilon^2,\rho^3\gamma^3\}}\right)$ iterations which also can be written as $\mathcal{O}((f(\mathbf{x}_0)-f(\mathbf{x}^*))(\epsilon^{-2}+\rho^{-3}\gamma^{-3}))$.

## 7.6 Proof of Theorem 2

In this proof, for notation convenience, we define $\epsilon'=\epsilon/2$ and $\gamma'=\gamma/2$.

First, note that the condition in Assumption 4 and the fact that $\nabla F(\mathbf{x},\boldsymbol{\theta})$ and $\nabla^2 F(\mathbf{x},\boldsymbol{\theta})$ are the unbiased estimators of the gradient $\nabla f(\mathbf{x})$ and Hessian $\nabla^2 f(\mathbf{x})$ imply that the variance of the batch gradient $\mathbf{d}_t$ and the batch Hessian $\mathbf{H}_t$ approximations are upper bounded by

$$\mathbb{E}\left[\|\mathbf{d}_t-\nabla f(\mathbf{x}_t)\|^2\right] \leq \frac{\nu^2}{b_g}, \qquad \mathbb{E}\left[\|\mathbf{H}_t-\nabla^2 f(\mathbf{x}_t)\|^2\right] \leq \frac{\xi^2}{b_H}. \tag{53}$$

Here we assume that $b_g$ and $b_H$ satisfy the following conditions,

$$b_g=\max\left\{\frac{324\nu^2 M^2 D^8}{\rho^4\gamma'^4},\frac{16D^2\nu^2}{\epsilon'^2}\right\}, \qquad b_H=\frac{81D^4\xi^2}{\rho^2\gamma'^2}. \tag{54}$$

We further set the parameter $r$ as

$$r=\frac{\rho^2\gamma'^2}{18MD^3}. \tag{55}$$

Now we proceed to analyze the complexity of Algorithm 2. First, consider the case that the current iterate $\mathbf{x}_t$ satisfies the inequality $\mathbf{d}_t^\top(\mathbf{v}_t-\mathbf{x}_t)<-\epsilon'$ and therefore we perform the first-order update in step 4. In this case, we can show that

$$\begin{aligned}
f(\mathbf{x}_{t+1}) &\leq f(\mathbf{x}_t)+\nabla f(\mathbf{x}_t)^\top(\mathbf{x}_{t+1}-\mathbf{x}_t)+\frac{L}{2}\|\mathbf{x}_{t+1}-\mathbf{x}_t\|^2 \\
&= f(\mathbf{x}_t)+\eta\nabla f(\mathbf{x}_t)^\top(\mathbf{v}_t-\mathbf{x}_t)+\frac{\eta^2 L}{2}\|\mathbf{v}_t-\mathbf{x}_t\|^2 \\
&\leq f(\mathbf{x}_t)+\eta\mathbf{d}_t^\top(\mathbf{v}_t-\mathbf{x}_t)+\eta(\nabla f(\mathbf{x}_t)-\mathbf{d}_t)^\top(\mathbf{v}_t-\mathbf{x}_t)+\frac{\eta^2 LD^2}{2} \\
&\leq f(\mathbf{x}_t)-\eta\epsilon'+\eta D\|\nabla f(\mathbf{x}_t)-\mathbf{d}_t\|+\frac{\eta^2 LD^2}{2},
\end{aligned} \tag{56}$$

where in the last inequality we used $\mathbf{d}_t^\top(\mathbf{v}_t-\mathbf{x}_t)<-\epsilon'$ and the fact that both $\mathbf{v}_t$ and $\mathbf{x}_t$ belong to the set $\mathcal{C}$ and therefore $\|\mathbf{x}_t-\mathbf{v}_t\|\leq D$. Consider $\mathcal{F}_t$ as the sigma algebra that measures all sources of randomness up to step $t$. Then, computing the expected value of both sides of (56) given $\mathcal{F}_t$ leads to

$$\mathbb{E}\left[f(\mathbf{x}_{t+1})\mid\mathcal{F}_t\right]\leq f(\mathbf{x}_t)-\eta\epsilon'+\frac{\eta D\nu}{\sqrt{b_g}}+\frac{\eta^2 LD^2}{2} \tag{57}$$

where we used the inequality $\mathbb{E}\left[X\right]\leq\sqrt{\mathbb{E}\left[X^2\right]}$ when $X$ is a positive random variable. Replace the stepsize $\eta$ by its value $\epsilon'/(D^2 L)$ and the batch size $b_g$ by its lower bound $(16D^2\nu^2)/(\epsilon'^2)$ to obtain

$$\mathbb{E}\left[f(\mathbf{x}_{t+1})\mid\mathcal{F}_t\right]\leq f(\mathbf{x}_t)-\frac{\epsilon'^2}{4D^2 L}. \tag{58}$$

Hence, in this case, the objective function value decreases in expectation by a constant factor of $\mathcal{O}(\epsilon'^2)$.

Now we proceed to study the case that the current iterate $\mathbf{x}_t$ does not satisfy the inequality $\mathbf{d}_t^\top (\mathbf{v}_t - \mathbf{x}_t) < -\epsilon'$ and we need to perform the second-order update in step 8. In this case, we can show that

$$
\begin{aligned}
f(\mathbf{x}_{t+1}) &\leq f(\mathbf{x}_t) + \nabla f(\mathbf{x}_t)^\top (\mathbf{x}_{t+1} - \mathbf{x}_t) + \frac{1}{2}(\mathbf{x}_{t+1} - \mathbf{x}_t)^\top \nabla^2 f(\mathbf{x})(\mathbf{x}_{t+1} - \mathbf{x}_t) + \frac{M}{6}\|\mathbf{x}_{t+1} - \mathbf{x}_t\|^3 \\
&\leq f(\mathbf{x}_t) + \sigma \nabla f(\mathbf{x}_t)^\top (\mathbf{u}_t - \mathbf{x}_t) + \frac{\sigma^2}{2}(\mathbf{u}_t - \mathbf{x}_t)^\top \nabla^2 f(\mathbf{x})(\mathbf{u}_t - \mathbf{x}_t) + \frac{\sigma^3 M D^3}{6} \\
&\leq f(\mathbf{x}_t) + \sigma \mathbf{d}_t^\top (\mathbf{u}_t - \mathbf{x}_t) + \sigma(\nabla f(\mathbf{x}_t) - \mathbf{d}_t)^\top (\mathbf{u}_t - \mathbf{x}_t) + \frac{\sigma^2}{2}(\mathbf{u}_t - \mathbf{x}_t)^\top \mathbf{H}_t(\mathbf{u}_t - \mathbf{x}_t) \\
&\quad + \frac{\sigma^2}{2}(\mathbf{u}_t - \mathbf{x}_t)^\top (\nabla^2 f(\mathbf{x}) - \mathbf{H}_t)(\mathbf{u}_t - \mathbf{x}_t) + \frac{\sigma^3 M D^3}{6}.
\end{aligned}
\tag{59}
$$

Note that $\mathbf{u}_t$ is a $\rho$-approximate solution for the subproblem in step 6 of Algorithm 2, with the objective function value less than $-\rho\gamma'$. This observation implies that the quadratic term $(\mathbf{u}_t - \mathbf{x}_t)^\top \mathbf{H}_t(\mathbf{u}_t - \mathbf{x}_t)$ is bounded above by $-\rho\gamma'$. Further, the linear term $\mathbf{d}_t^\top (\mathbf{u}_t - \mathbf{x}_t)$ is less than $r$ according to the constraint of the subproblem. Applying these substitutions and using the Cauchy-Schwartz inequality multiple times lead to

$$
f(\mathbf{x}_{t+1}) \leq f(\mathbf{x}_t) + \sigma r + \sigma D\|\mathbf{d}_t - \nabla f(\mathbf{x}_t)\| - \frac{\sigma^2 \rho\gamma'}{2} + \frac{\sigma^2 D^2}{2}\|\mathbf{H}_t - \nabla^2 f(\mathbf{x})\| + \frac{\sigma^3 M D^3}{6}.
\tag{60}
$$

Compute the conditional expected value of both sides of (60) and use the inequalities in (53) to obtain

$$
\mathbb{E}\left[f(\mathbf{x}_{t+1}) \mid \mathcal{F}_t\right] \leq f(\mathbf{x}_t) + \sigma r + \frac{\sigma D \nu}{\sqrt{b_g}} - \frac{\sigma^2 \rho\gamma'}{2} + \frac{\sigma^2 D^2 \xi}{2\sqrt{b_H}} + \frac{\sigma^3 M D^3}{6}.
\tag{61}
$$

By setting the stepsize $\sigma = \rho\gamma'/MD^3$ in (61) it follows that

$$
\mathbb{E}\left[f(\mathbf{x}_{t+1}) \mid \mathcal{F}_t\right] \leq f(\mathbf{x}_t) - \frac{\rho^3 \gamma'^3}{3L^2 D^6} + \frac{r\rho\gamma'}{MD^3} + \frac{\rho\gamma'\nu}{MD^2 \sqrt{b_g}} + \frac{\rho^2 \gamma'^2 \xi}{2M^2 D^4 \sqrt{b_H}}.
\tag{62}
$$

Moreover, setting $r = \frac{\rho^2 \gamma'^2}{18MD^3}$ and $b_H = \frac{81 D^4 \xi^2}{\rho^2 \gamma'^2}$, and replacing $b_g$ by its lower bound $\frac{324\nu^2 M^2 D^8}{\rho^4 \gamma'^4}$ lead to

$$
\mathbb{E}\left[f(\mathbf{x}_{t+1}) \mid \mathcal{F}_t\right] \leq f(\mathbf{x}_t) - \frac{\rho^3 \gamma'^3}{6M^2 D^6}
\tag{63}
$$

Hence, in this case, the expected objective function value decreases by a constant of $\mathcal{O}(\rho^3 \gamma'^3)$.

By combining the results in (58) and (63), we obtain that if the iterate $\mathbf{x}_t$ is not the final iterate the objective function value at step $t + 1$ satisfies the following ineqaulity

$$
\mathbb{E}\left[f(\mathbf{x}_{t+1}) \mid \mathcal{F}_t\right] \leq f(\mathbf{x}_t) - \min\left\{\frac{\epsilon'^2}{4LD^2}, \frac{\rho^3 \gamma'^3}{6M^2 D^6}\right\}.
\tag{64}
$$

Let us define $T$ as the number of iterations we perform until Algorithm 2 stops. We use an argument similar to Wald's lemma to derive an upper bound on the expected number of iterations $T$ that we

need to run the algorithm. Note that

$$\mathbb{E}\left[f(\mathbf{x}_0) - f(\mathbf{x}_T)\right] = \mathbb{E}\left[\sum_{t=1}^{T}(f(\mathbf{x}_{t-1}) - f(\mathbf{x}_t))\right]$$

$$= \mathbb{E}\left[\mathbb{E}\left[\sum_{t=1}^{T}(f(\mathbf{x}_{t-1}) - f(\mathbf{x}_t))\right]\bigg|\, T\right]$$

$$= \sum_{k=1}^{\infty}\mathbb{E}\left[\sum_{t=1}^{k}(f(\mathbf{x}_{t-1}) - f(\mathbf{x}_t))\right]\mathbb{P}(T = k)$$

$$= \sum_{k=1}^{\infty}\sum_{t=1}^{k}\mathbb{E}\left[(f(\mathbf{x}_{t-1}) - f(\mathbf{x}_t))\right]\mathbb{P}(T = k)$$

$$\geq \sum_{k=1}^{\infty}\sum_{t=1}^{k}\min\left\{\frac{\epsilon'^2}{4LD^2}, \frac{\rho^3\gamma'^3}{6M^2D^6}\right\}\mathbb{P}(T = k)$$

$$= \min\left\{\frac{\epsilon'^2}{4LD^2}, \frac{\rho^3\gamma'^3}{6M^2D^6}\right\}\sum_{k=1}^{\infty}k\,\mathbb{P}(T = k)$$

$$= \min\left\{\frac{\epsilon'^2}{4LD^2}, \frac{\rho^3\gamma'^3}{6M^2D^6}\right\}\mathbb{E}\left[T\right]. \tag{65}$$

The first equality holds by simplifying the sum, for the second equality we use the fact that $\mathbb{E}\left[X\right] = \mathbb{E}\left[\mathbb{E}\left[X \mid Y\right]\right]$, in the third equality we use the expression $\mathbb{E}\left[\mathbb{E}\left[X \mid Y\right]\right] = \sum_y \mathbb{E}\left[X \mid Y = y\right]\mathbb{P}(Y = y)$, in the fourth equality we exchange sum and expectation, and the inequality is true based on the result in (64). Note that to derive this result we also have assumed that the sequence of function differences $f(\mathbf{x}_{t-1}) - f(\mathbf{x}_t)$ are independent of each other and also independent of the total number of iterations $T$.

Based on the result in (65), we can write that $\mathbb{E}\left[T\right] \leq \mathbb{E}\left[f(\mathbf{x}_0) - f(\mathbf{x}_T)\right]/\min\left\{\frac{\epsilon'^2}{4LD^2}, \frac{\rho^3\gamma'^3}{6M^2D^6}\right\}$. We further know that $f(\mathbf{x}_T) \geq f(\mathbf{x}^*)$ which implies that

$$\mathbb{E}\left[T\right] \leq (f(\mathbf{x}_0) - f(\mathbf{x}^*))\max\left\{\frac{4LD^2}{\epsilon'^2}, \frac{6M^2D^6}{\rho^3\gamma'^3}\right\}. \tag{66}$$

Using Markov's inequality we can show that

$$\mathbb{P}\left(T \leq a\right) \geq 1 - \frac{(f(\mathbf{x}_0) - f(\mathbf{x}^*))\max\left\{\frac{4LD^2}{\epsilon'^2}, \frac{6M^2D^6}{\rho^3\gamma'^3}\right\}}{a} \tag{67}$$

Set $a = \frac{(f(\mathbf{x}_0) - f(\mathbf{x}^*))}{\delta}\max\left\{\frac{4LD^2}{\epsilon'^2}, \frac{6M^2D^6}{\rho^3\gamma'^3}\right\}$ to obtain that

$$\mathbb{P}\left(T \leq \frac{(f(\mathbf{x}_0) - f(\mathbf{x}^*))\max\left\{\frac{4LD^2}{\epsilon'^2}, \frac{6M^2D^6}{\rho^3\gamma'^3}\right\}}{\delta}\right) \geq 1 - \delta. \tag{68}$$

Therefore, it follows that with high probability the total number of iterations $T$ that Algorithm 2 runs is at most $\mathcal{O}(\max\left\{\epsilon'^{-2}, \rho^{-3}\gamma'^{-3}\right\})$.

Now it remains to show that the outcome of Algorithm 2 is an $(\epsilon, \gamma)$-SOSP of Problem (33) with high probability. Let's assume that $\mathbf{x}_t$ is the final output of Algorithm 2. Then, we know that $\mathbf{x}_t$ satisfies the conditions

$$\mathbf{d}_t^{\top}(\mathbf{x} - \mathbf{x}_t) \geq -\epsilon' \quad \text{for all } \mathbf{x} \in \mathcal{C}, \tag{69}$$

and

$$(\mathbf{x} - \mathbf{x}_t)^{\top}\mathbf{H}_t(\mathbf{x} - \mathbf{x}_t) \geq -\gamma' \quad \text{for all } \mathbf{x} \in \mathcal{C},\ \mathbf{d}_t^{\top}(\mathbf{x} - \mathbf{x}_t) \leq r. \tag{70}$$

First, we use the condition in (69) to show that $\mathbf{x}_t$ satisfies the first-order optimality condition with high probability. Note that for any $\mathbf{x} \in \mathcal{C}$ it holds that

$$\nabla f(\mathbf{x}_t)^{\top}(\mathbf{x} - \mathbf{x}_t) = \mathbf{d}_t^{\top}(\mathbf{x} - \mathbf{x}_t) + (\nabla f(\mathbf{x}_t) - \mathbf{d}_t)^{\top}(\mathbf{x} - \mathbf{x}_t)$$

$$\geq \mathbf{d}_t^{\top}(\mathbf{x} - \mathbf{x}_t) - D\|\nabla f(\mathbf{x}_t) - \mathbf{d}_t\|. \tag{71}$$

Now compute the minimum of both sides of (71) for all $\mathbf{x} \in \mathcal{C}$ to obtain

$$
\begin{aligned}
\min_{\mathbf{x} \in \mathcal{C}} \{\nabla f(\mathbf{x}_t)^\top (\mathbf{x} - \mathbf{x}_t)\} &\geq \min_{\mathbf{x} \in \mathcal{C}} \{\mathbf{d}_t^\top (\mathbf{x} - \mathbf{x}_t) - D \|\nabla f(\mathbf{x}_t) - \mathbf{d}_t\|\} \\
&= \min_{\mathbf{x} \in \mathcal{C}} \{\mathbf{d}_t^\top (\mathbf{x} - \mathbf{x}_t)\} - D \|\nabla f(\mathbf{x}_t) - \mathbf{d}_t\| \\
&\geq -\epsilon' - D \|\nabla f(\mathbf{x}_t) - \mathbf{d}_t\|,
\end{aligned}
\tag{72}
$$

where the equality holds since $D \|\nabla f(\mathbf{x}_t) - \mathbf{d}_t\|$ does not depend on $\mathbf{x}$, and the last inequality is implied by (69). Since $\mathbb{E} \left[ \|\nabla f(\mathbf{x}_t) - \mathbf{d}_t\|^2 \right] \leq \nu^2/b_g$ we obtain from Markov's inequality that

$$
\mathbb{P} \left( \|\nabla f(\mathbf{x}_t) - \mathbf{d}_t\| \leq \epsilon'' \right) \geq 1 - \frac{\nu^2}{b_g \epsilon''^2}.
\tag{73}
$$

Therefore, by combining the results in (72) and (73) we obtain that

$$
\mathbb{P} \left( \min_{\mathbf{x} \in \mathcal{C}} \{\nabla f(\mathbf{x}_t)^\top (\mathbf{x} - \mathbf{x}_t)\} \geq -(\epsilon' + D\epsilon'') \right) \geq 1 - \frac{\nu^2}{b_g \epsilon''^2}.
\tag{74}
$$

Now by setting $\epsilon'' = \epsilon'/D$ it follows from (74) that with probability at least $1 - \nu^2 D^2/b_g \epsilon'^2$ the final iterate $\mathbf{x}_t$ satisfies

$$
\nabla f(\mathbf{x}_t)^\top (\mathbf{x} - \mathbf{x}_t) \geq -2\epsilon' \qquad \text{for all } \mathbf{x} \in \mathcal{C}.
\tag{75}
$$

Replacing $\epsilon'$ by $\epsilon/2$ leads to

$$
\nabla f(\mathbf{x}_t)^\top (\mathbf{x} - \mathbf{x}_t) \geq -\epsilon \qquad \text{for all } \mathbf{x} \in \mathcal{C}.
\tag{76}
$$

It remains to show that with high probability the final iterate $\mathbf{x}_t$ satisfies the second-order optimality condition.

First, consider the sets $\mathcal{A}_t = \{\mathbf{x} \mid \nabla f(\mathbf{x}_t)^\top (\mathbf{x} - \mathbf{x}_t) = 0\}$ and $\mathcal{B}_t = \{\mathbf{x} \mid \mathbf{d}_t^\top (\mathbf{x} - \mathbf{x}_t) \leq r\}$. We proceed to show that with high probability $\mathcal{A}_t \subset \mathcal{B}_t$. If $\mathbf{y}$ satisfies the condition

$$
\nabla f(\mathbf{x}_t)^\top (\mathbf{y} - \mathbf{x}_t) = 0,
\tag{77}
$$

then it can be shown that

$$
\begin{aligned}
\mathbf{d}_t^\top (\mathbf{y} - \mathbf{x}_t) &\leq \nabla f(\mathbf{x}_t)^\top (\mathbf{y} - \mathbf{x}_t) + (\mathbf{d}_t - \nabla f(\mathbf{x}_t))^\top (\mathbf{y} - \mathbf{x}_t) \\
&\leq D \|\mathbf{d}_t - \nabla f(\mathbf{x}_t)\|.
\end{aligned}
\tag{78}
$$

Since $\mathbb{E} \left[ \|\nabla f(\mathbf{x}_t) - \mathbf{d}_t\|^2 \right] \leq \nu^2/b_g$ we obtain from Markov's inequality that

$$
\mathbb{P} \left( \|\nabla f(\mathbf{x}_t) - \mathbf{d}_t\| \leq \frac{r}{D} \right) \geq 1 - \frac{\nu^2 D^2}{b_g r^2}.
\tag{79}
$$

Therefore, by combining the results in (78) and (79) we obtain that

$$
\mathbb{P} \left( \mathbf{d}_t^\top (\mathbf{y} - \mathbf{x}_t) \leq r \right) \geq 1 - \frac{\nu^2 D^2}{b_g r^2}.
\tag{80}
$$

This argument shows that if $\mathbf{y} \in \mathcal{A}_t$, then it also belongs to the set $\mathcal{B}_t$, i.e., $\mathbf{y} \in \mathcal{B}_t$, with high probability. This result shows if an inequality holds for all $\mathbf{x}$ that satisfy $\mathbf{d}_t^\top (\mathbf{x} - \mathbf{x}_t) \leq r$, then with high probability that inequality also holds for all $\mathbf{x}$ that satisfy the condition $\nabla f(\mathbf{x}_t)^\top (\mathbf{x} - \mathbf{x}_t) = 0$.

Now, note that if $\mathbf{x}_t$ is the output of Algorithm 2, then for any $\mathbf{x} \in \mathcal{C}$ satisfying $\mathbf{d}_t^\top (\mathbf{x} - \mathbf{x}_t) \leq r$ it holds that

$$
\begin{aligned}
(\mathbf{x} - \mathbf{x}_t)^\top \nabla^2 f(\mathbf{x}_t)(\mathbf{x} - \mathbf{x}_t) &= (\mathbf{x} - \mathbf{x}_t)^\top \mathbf{H}_t (\mathbf{x} - \mathbf{x}_t) - (\mathbf{x} - \mathbf{x}_t)^\top (\mathbf{H}_t - \nabla^2 f(\mathbf{x}_t))(\mathbf{x} - \mathbf{x}_t) \\
&\geq -\gamma' - D^2 \|\mathbf{H}_t - \nabla^2 f(\mathbf{x}_t)\|.
\end{aligned}
\tag{81}
$$

Further, define the random variable $X_t = \|\mathbf{H}_t - \nabla^2 f(\mathbf{x}_t)\|$. As we know that $\mathbb{E} \left[ X_t^2 \right] \leq \xi^2/b_H$, it follows by Markov's inequality that $\mathbb{P}(X_t \leq a) \geq 1 - \xi^2/(b_H a^2)$. Therefore, we can write that

$$
\mathbb{P}(\|\mathbf{H}_t - \nabla^2 f(\mathbf{x}_t)\| \leq \gamma'') \geq 1 - \frac{\xi^2}{b_H \gamma''^2}.
\tag{82}
$$

Hence, by using the results in (81) and (82), we can show that with probability at least $1 - \frac{\xi^2}{b_H \gamma''^2}$ for any $\mathbf{x} \in \mathcal{C}$ satisfying $\mathbf{d}_t^\top (\mathbf{x} - \mathbf{x}_t) \leq r$ it holds

$$(\mathbf{x} - \mathbf{x}_t)^\top \nabla^2 f(\mathbf{x}_t)(\mathbf{x} - \mathbf{x}_t) \geq -\gamma' - D^2 \gamma''. \tag{83}$$

By setting $\gamma'' = \gamma'/D^2$ it follows that $\mathbf{x}_t$ satisfies the condition

$$(\mathbf{x} - \mathbf{x}_t)^\top \nabla^2 f(\mathbf{x}_t)(\mathbf{x} - \mathbf{x}_t) \geq -2\gamma' \quad \text{for all } \mathbf{x} \in \mathcal{C}, \mathbf{d}_t^\top (\mathbf{x} - \mathbf{x}_t) \leq r, \tag{84}$$

with a probability larger than $1 - \frac{\xi^2 D^4}{b_H \gamma'^2}$. Further, with probability at least $1 - \frac{\nu^2 D^2}{b_g r^2}$ we know that $\mathcal{A}_t \subset \mathcal{B}_t$. These observations imply that if $\mathbf{x}_t$ is the output of Algorithm 2 it satisfies

$$(\mathbf{x} - \mathbf{x}_t)^\top \nabla^2 f(\mathbf{x}_t)(\mathbf{x} - \mathbf{x}_t) \geq -2\gamma' \quad \text{for all } \mathbf{x} \in \mathcal{C}, \nabla f(\mathbf{x}_t)^\top (\mathbf{x} - \mathbf{x}_t) = 0, \tag{85}$$

with probability at least $1 - \frac{\xi^2 D^4}{b_H \gamma'^2} - \frac{\nu^2 D^2}{b_g r^2}$, where we used the inequality

$$\begin{aligned} P(A \cap B) &= P(A) + P(B) - P(A \cup B) \\ &\geq P(A) + P(B) - 1. \end{aligned} \tag{86}$$

By setting $\gamma' = \gamma/2$ we obtain that with probability at least $1 - \frac{\xi^2 D^4}{b_H \gamma'^2} - \frac{\nu^2 D^2}{b_g r^2}$ the final iterate satisfies the condition

$$(\mathbf{x} - \mathbf{x}_t)^\top \nabla^2 f(\mathbf{x}_t)(\mathbf{x} - \mathbf{x}_t) \geq -\gamma \quad \text{for all } \mathbf{x} \in \mathcal{C}, \nabla f(\mathbf{x}_t)^\top (\mathbf{x} - \mathbf{x}_t) = 0. \tag{87}$$

Therefore, with probability at least $1 - \frac{\nu^2 D^2}{b_g \epsilon'^2} - \frac{\xi^2 D^4}{b_H \gamma'^2} - \frac{\nu^2 D^2}{b_g r^2}$ the output of Algorithm 2 is an $(\epsilon, \gamma)$-SOSP of the stochastic optimization problem in (33). This observation and the conditions on the batch sizes in (54) implies that the output of Algorithm 2 is an $(\epsilon, \gamma)$-SOSP of the stochastic optimization problem in (33) with probability at least $1 - \frac{1}{16} - \frac{1}{324} - \frac{\rho^2}{81} \geq 0.92$. (Note that $\rho \leq 1$). Indeed, by increasing the size of batches $b_g$ and $b_H$ all the results hold with a higher probability.