[Reviews · NeurIPS 2018]

Reviewer 1



In this submission, the authors propose an algorithm for nonconvex optimization under convex constraints. The method aims at escaping saddle points and converging to second-order stationary points. The main idea of the submission lies in exploiting the structure of the convex constraint set so as to find an approximate solution to quadratic programs over this set in polynomial time. By combining this oracle with a first-order iteration such as conditional gradient or projected gradient, the authors are able to guarantee that their algorithm will converge towards an $(\epsilon,\gamma)$-Second-Order Stationary Point (SOSP), i. e. a point at which first and second-order necessary conditions are approximately satisfied, respectively to $\epsilon$ and $\gamma$ tolerances. In addition to addressing a topic of increasing interest in the community of NIPS (escaping saddle points in nonconvex optimization), this submission also touches complexity aspects that are of general interest in theoretical computer science (NP-hardness and approximation algorithms). The authors clearly introduce the problem and their objectives, while positioning it within the existing literature. My main comment about the introductory section is that this submission claims that it provides the first finite-time complexity analysis for reaching an SOSP" of the considered problem. I believe this is not entirely true, considering the following work: C. Cartis, N. I. M. Gould and Ph. L. Toint, Second-Order Optimality and Beyond: Characterization and Evaluation Complexity in Convexly Constrained Nonlinear Optimization", Foundations of Computational Mathematics, DOI 10.1007/s10208-017-9363-y. In this reference, the authors derive a bound in $\epsilon^{-3}$, in terms of outer iterations and function/derivatives evaluations, for obtaining a point at which approximate second-order conditions are satisfied. Although their optimality measure differs from that of this submission, I believe it is clear that their method also escapes saddle points and is thus worth mentioning (as mentioned by the authors regarding other references, the cost of solving the subproblem is not explicitly taken into account). Their optimality measure only involves one tolerance, yet one can view their definition of first-order optimality as similar to that of the submission, and I would welcome the authors' comments on this reference. I would have expected Definitions 2 and 3 to enforce feasibility of the point in addition to the other optimality conditions. In the present submission, this is not a critical issue as all considered iterates are computed so as to be feasible, however I believe adding feasible" in these definitions would make more sense, because it does not seem that infeasible points are desirable in the context of this submission. I would like to have the authors' view on this point. The presentation of the main algorithm and result in Section 3 is in my opinion well conducted. Even though it is clear from the algorithmic process, the authors could have insisted more on the fact that the method can have cheap first-order iterations, which is one of the strengths of this framework. The description of the first-order steps and algorithms in Section 4 is extremely clear. Section 5 provides an example of the second phase of the algorithm (second-order step) when the constraint set is defined as an intersection of ellipsoids. Using a result from approximation algorithms, the authors show that in that case, it is possible to obtain a solution sufficiently close to the global optimum, thereby enabling the derivation of a complexity bound. Another example for linear constraints is provided in appendix. One might be concerned with the applicability of such results, especially given their dependencies on the problem dimension. Still, the argument appears valid, and the considered cases are among the most commonly encountered. The last technical section of the submission, Section 6, is concerned with extending the previous results to the case where the gradient and Hessian of the objective are only accessed through stochastic estimates. Using batch gradients and Hessians with appropriate batch sizes, it is possible to obtain complexity guarantees for reaching approximate second-order stationary points with high probability. This extension and the associated algorithmic modifications are very interesting: in particular, the relaxation of the constraint $d_t^\top (u-x_t)=0$ is a key feature of the stochastic method, which is briefly but clearly discussed by the authors. It seems that the condition on Line 3 of Algorithm 2 is also a key difference, about which a comment could have been made. In addition, the first-order-type step is computed at every iteration, probably as a consequence of the stochasticity: this is a key difference in the per-iteration cost of the method, even though it does not impact the complexity analysis. I acknowledge that the authors could not add an extensive discussion due to space constraints, but part of it could have been added as supplementary material. In a nutshell, I believe the results of the submission to be correct, clearly introduced and motivated. Although I regret the lack of numerical experiments that would have illustrated the applicability of the proposed framework, it is my opinion that this submission can still have an impact on the community, and stimulate further work on algorithms of this form. Therefore, I will give this submission an overall score of 8. Additional comments: a) There are a few minor typos, e.g. "propsoed" line 43. b) The series of equations on line 523 page 15 is an interesting derivation, but the first lines might need clarification. More precisely, I would like the authors to comment upon the two expectation operators that appear in the first three equalities, and the argument for going from the third equality to the fourth one. Edit following author feedback: I thank the authors for providing detailed answers to my comments, and I trust them to incorporate revisions as stated. I agree with the other reviewers' comments regarding the ability of the proposed scheme to escape strict saddle points, but I am confident that this can be addressed by the authors. As a result, I maintain my initial assessment on the submission.

Reviewer 2



The paper proposes a method (based on Frank-Wolfe or projected gradient) that, using second-order information (Hessian), escapes saddle points of a smooth non-convex optimization problem subject to a convex set $\mathcal{C}$. The results rely on the feasibility of finding an approximate solution to a quadratic program in polynomial time. Under this requirement, and reasonable assumptions, the method is shown to converge to an approximate second-order stationary point. The results are also extended to a stochastic setting. Convergence to a second-order stationary point of first order methods was previously studied in the unconstrained case. This paper is one of the first (or probably the first) to consider a convex constraint case, in the particular case where the minimization of a quadratic loss over such set is feasible. The results of this paper are interesting and correct to the best of my knowledge. Thus, I recommend the paper for publication, provided the authors improve the manuscript as described below. 0. The results of this paper are only proved for projected gradient or Frank-Wolfe methods. Therefore, the authors should not claim that they provide a "general framework" for finding second order stationary points. I recommend that this should be corrected throughout the paper. 1. Overall, I suggest a thorough proof read of the text. 2. In the entire paper, replace "an SOSP" to "a SOSP". 3. line 43: Proposoed 4. line 44: "advances in constant-factor optimization for nonconvex quadratic programs": a citation is appropriate. Which advances? Based on which works? These are also not cited in related work. 5. line 92: arbitrary small -> arbitrarily small 6. Proposition 1 is well-known and trivial. The author should cite a reference in the statement, such as Bertsekas 1999, and call it a Theorem instead of Proposition. 7. Why stating the definition of strict saddle (Definition 1) if this is nowhere used? Maybe citing a reference is enough for the unconstrained case, and leave Definition 3 for clarity. 8. Definition 2. I suggest starting the statement as ... Let $f: \mathbb{R} \to \mathbb{R}$ be a twice continuously differentiable function, and let $C \subset \mathbb{R}^d$ be closed and convex. Then, $x^*$ is called an ... 9. line 145: I.e. should be ", i.e." with a comma not a period after FOSP. This should also be corrected in several other parts of the manuscript. 10. line 154: why using OPT instead of simply $q(u^*)$ to denote the optimal objective value. 11. line 261: use Quadratically Constrained Quadratic Program (QCQP) 12. Replace Proposition 5 by Theorem 5 ===== I apologize in suggesting the authors to replace "an SOSP" to "a SOSP", which may be wrong. Please, keep the original "an SOSP" or double check which one is correct. It is kind of confusing because if you read the abbreviation how it sounds then you should use "an" because of the "S -> Ess", however, without abbreviation one would say "a Second Order Stationary Point". Thus, I'm still confused which way is correct. Maybe both?

Reviewer 3



The authors address the problem of escaping from saddle points in smooth nonconvex optimization problems constrained by a convex set C. A framework is proposed that allows iterates to converge to an (eps, gamma)-SOSP in a finite number of steps. First, an eps-FOSP stationary point is found using an appropriate first-order method (the authors suggest Frank-Wolfe or projected gradient descent). This stationary point may be a saddle point or a local maximizer, so as a second step, a constrained quadratic program is solved to find an (eps, gamma)-SOSP of the original problem. In addition to assuming the constraint set C is convex, the framework also assumes that a constant stepsize is used and that the necessary QPs can be solved in a finite number of arithmetic iterations. The authors provide an upper iteration bound, a lower bound for the decrease in the objective value for each of the first-order methods, and an extension to stochastic problems when the objective function is defined as an expectation of a set of stochastic functions. This appears to be a solid contribution, giving a sensible framework for tackling optimization problems where saddle points could be encountered by the first-order method, although I am unsure if saddle points will necessarily be escaped from using the framework (see below). The theoretical results appear to be correct, although I was unable to verify all of the proofs due to time constraints. While there are a number of assumptions that are made that enable the theoretical analysis, most (but not all) of these seem sensible in practical settings. However, I have a couple of concerns/comments/questions: - The framework does not require that the saddle points are delta-strict (line 121), which implies that the (eps, gamma)-SOSP could still be a saddle point. If this were to happen, what would be the best method to get out of this saddle point? Decrease gamma and try again? It does not seem like this framework actually guarantees an escape from saddle points, unless one decreases gamma to be small enough. - The assumption that the stepsize eta is constant is unrealistic since most practitioners decrease the stepsize as iterates get closer to the solution. How difficult is it to adapt the analysis to the case of variable stepsizes? - How much of the analysis can be applied if stochastic gradient descent (SGD) is used to solve the first-order problem? - The following sentence in the abstract is confusing: "We propose a generic framework that yields convergence to a second-order stationary point of the problem, if the convex set C is simple for a quadratic objective function". So the objective function needs to be quadratic? Because the text makes it seem like we are minimizing over a general smooth nonconvex function. - It would have been nice if there could have been some computational example of the method in action, but I realize there is a lack of space. Some editorial comments: - Line 32: "... later below)." Either "later" or "below", not both. - Line 34: semi-colon should be a colon. - Lin 43: "proposed". - Line 123: "... approximate local minimum". - Line 145: A sentence should preferably not be started with "i.e.", rather use "In other words" or something along those lines.